# Differential Transcript Profiles in Cumulus-Oocyte Complexes Originating from Pre-Ovulatory Follicles of Varied Physiological Maturity in Beef Cows

**DOI:** 10.3390/genes12060893

**Published:** 2021-06-10

**Authors:** Sarah E. Moorey, Jenna M. Monnig, Michael F. Smith, M. Sofia Ortega, Jonathan A. Green, Ky G. Pohler, G. Alan Bridges, Susanta K. Behura, Thomas W. Geary

**Affiliations:** 1Department of Animal Science, University of Tennessee, Knoxville, TN 37996, USA; 2Division of Animal Sciences, University of Missouri, Columbia, MO 65211, USA; monnigjm@missouri.edu (J.M.M.); smithmf@missouri.edu (M.F.S.); ortegaobandom@missouri.edu (M.S.O.); greenjo@missouri.edu (J.A.G.); behuras@missouri.edu (S.K.B.); 3Department of Animal Science, Texas A&M University, College Station, TX 77843, USA; kpohler@tamu.edu; 4Department of Animal Science, University of Minnesota, St. Paul, MN 55108, USA; gabridges@elanco.com; 5USDA-ARS Fort Keogh Livestock and Range Research Lab, Miles City, MT 59301, USA; tom.geary@usda.gov

**Keywords:** Cumulus-Oocyte complex, pre-ovulatory follicle, transcriptome, beef cow, induced ovulation

## Abstract

Small dominant follicle diameter at induced ovulation, but not at spontaneous ovulation, decreased pregnancy rate, fertilization rate, and day seven embryo quality in beef cows. We hypothesized that the physiological status of the follicle at GnRH-induced ovulation has a direct effect on the transcriptome of the Cumulus-Oocyte complex, thereby affecting oocyte competence and subsequent embryo development. The objective of this study was to determine if the transcriptome of oocytes and associated cumulus cells (CC) differed among small (≤11.7 mm) and large follicles (≥12.7 mm) exposed to a GnRH-induced gonadotropin surge and follicles (11.7–14.0 mm) exposed to an endogenous gonadotropin surge (spontaneous follicles). RNA sequencing data, from pools of four oocytes or their corresponding CC, revealed 69, 94, and 83 differentially expressed gene transcripts (DEG) among oocyte pools from small versus large, small versus spontaneous, and large versus spontaneous follicle classifications, respectively. An additional 128, 98, and 80 DEG were identified among small versus large, small versus spontaneous, and large versus spontaneous follicle CC pools, respectively. The biological pathway “oxidative phosphorylation” was significantly enriched with DEG from small versus spontaneous follicle oocyte pools (FDR < 0.01); whereas the glycolytic pathway was significantly enriched with DEG from CC pools obtained from large versus small follicles (FDR < 0.01). These findings collectively suggest that altered carbohydrate metabolism within the Cumulus-Oocyte complex likely contributes to the decreased competency of oocytes from small pre-ovulatory follicles exposed to an exogenous GnRH-induced gonadotropin surge.

## 1. Introduction

Acquisition of oocyte competence is essential for fertilization, embryo cleavage, and subsequent embryonic/fetal development [1]. Oocytes of high developmental competence have undergone meiotic, cytoplasmic, and molecular maturation to allow for resumption of meiosis, fertilization, appropriate cytoplasmic organelle function, and accumulation of maternal messenger ribonucleic acid (mRNA) transcripts for early embryonic development [2,3,4]. Although bovine oocytes enclosed in follicles ≥3 mm in diameter have the ability to resume meiosis (i.e., nuclear maturation; [5]) acquisition of oocyte competence continues until germinal vesicle breakdown is initiated by the pre-ovulatory gonadotropin surge [6]. During the final stages of oocyte maturation (oocyte capacitation; [7]), organelles are redistributed to achieve optimal developmental capacity. More specifically, oocyte mitochondrial number and distribution are modified to enhance adenosine triphosphate (ATP) production via oxidative phosphorylation [8].

During folliculogenesis, cumulus cells (CC) transfer critical metabolic substrates, such as pyruvate, to the oocyte for ATP production [9,10]. Furthermore, the surrounding CC contribute to the oocyte transcriptome through the transfer of mRNA to the oocyte through trans-zonal processes that span the zona pellucida and form direct connections between the cumulus cells and the oolemma [11,12]. Trans-zonal connections remain intact during folliculogenesis until the pre-ovulatory gonadotropin surge, at which time they are disrupted and the transfer of vital substrates ceases [13]. At the secondary follicle stage and continuing throughout growth of the oocyte, transcripts of mRNA are synthesized and stored for utilization by the oocyte and early embryo until activation of the embryonic genome [14,15,16,17].

Although a positive correlation exists between antral follicle size and oocyte developmental competence in small to medium-sized antral follicles in cattle, far less is known about the association between the physiological maturity of an ovulatory follicle and acquisition of oocyte competence. Female cattle recruit two to three waves of multiple antral follicles for growth during the estrous cycle, and one follicle becomes dominant during each follicular wave [6]. While progesterone levels are high, the dominant follicle cannot continue growth and will regress to allow the onset of a new follicular wave. The dominant follicle from the second or third follicular wave continues growth during the reduced progesterone levels that follow luteolysis. This follicle ovulates following the pre-ovulatory gonadotropin surge, and its physiological status near the time of ovulation may affect establishment and maintenance of pregnancy by affecting oocyte competence and the maternal environment [18,19]. In postpartum beef cows, gonadotropin releasing hormone (GnRH) injection to induce ovulation of small dominant follicles (<11.3 mm) resulted in reduced pregnancy rates and an increased incidence of late embryonic mortality [20]. There is compelling evidence that the lowered pregnancy rate following GnRH-induced ovulation of small dominant follicles results from a compromised maternal environment, due to lower circulating concentrations of pre-ovulatory estradiol and post-ovulatory progesterone [18,19,21].

Furthermore, acquisition of oocyte developmental competence may not be complete when small dominant follicles are induced to ovulate, since fertilization rate and embryo quality were decreased on day seven following artificial insemination and GnRH-induced ovulation of small compared to large dominant bovine follicles [18]. An exogenous GnRH-induced gonadotropin surge, in a cow with a physiologically immature follicle, may induce a premature stop to oocyte transcription and (or) transfer of mRNA and energy substrates from the cumulus cells to the oocyte and thereby reduce the competence of an oocyte for fertilization and further embryonic development. We hypothesized that differentially expressed gene transcripts (DEG) of oocytes or CC among small and large or spontaneous follicles would reveal potential physiological processes that may be compromised when small follicles undergo GnRH-induced ovulation. Although previous studies have identified granulosa and CC markers of oocyte competence [22,23,24,25,26,27,28,29], we further hypothesized that investigating the effect of the physiological status of a pre-ovulatory follicle size on the oocyte and cumulus cell transcriptomes would reveal additional transcripts and biological pathways, associated with the final stages of folliculogenesis, that may be important for acquisition of oocyte maturity. To test these hypotheses, RNA sequencing analysis was performed to determine if the transcriptome of oocytes and associated CC differed following an exogenous GnRH-induced gonadotropin surge in cows having small (≤11.7 mm; no estrus expression) or large (≥12.7 mm; no estrus expression) follicles as well as follicles that result in estrus expression and an endogenous gonadotropin surge (spontaneous follicles; 11.7–14.0 mm; estrus expressed; no GnRH).

## 2. Materials and Methods

All protocols and procedures were approved by the Fort Keogh Livestock and Range Research Laboratory Animal Care and Use Committee (IACUC approval number 022014-2).

### 2.1. Animal Procedures and Synchronization of Ovulation

A timeline for synchronization of ovulation, blood collection, ovarian mapping of follicles and corpora lutea, and transvaginal aspiration is depicted in Figure 1a. Estrous cycles of suckled postpartum beef cows (Hereford-Angus crossbred; *n* = 250), ages 4–13 years (mean = 6), were pre-synchronized using a protocol that consisted of administration of GnRH (i.m.; 100 µg; Factrel^®^; Zoetis Animal Health, Kalamazoo, MI, USA) and insertion of a CIDR (intravaginal insert; 1.38 g progesterone; Eazi-Breed CIDR^®^; Zoetis Animal Health, Kalamazoo, MI, USA). Five days after CIDR insertion and GnRH injection, the CIDR was removed and all animals received two injections of prostaglandin F2α (PGF; i.m.; 25 mg; Lutalyse^®^, Zoetis Animal Health, Kalamazoo, MI, USA). Synchronization was applied to cows in five replicates to facilitate transvaginal aspiration, with 50 cows per replicate. Ten to fourteen days after pre-synchronization, cows received an injection of GnRH (GnRH1; i.m.; 100 µg; Factrel^®^) on day −9 to initiate a new follicular wave (Figure 1a). On day −2, cows received an injection of PGF (i.m.; 25 mg; Lutalyse^®^) to lyse any corpora lutea, and on day 0, a second injection of GnRH (GnRH2; i.m.; 100 µg; Factrel^®^) was administered to induce a pre-ovulatory gonadotropin surge in cows that had not expressed estrus.

On day −9, cows received an estrous detection patch (Estrotect^®^; Rockway Inc.; Spring Valley, WI, USA) that was monitored daily from day −9 to −2 and replaced if scratched. Visual assessment of estrous expression combined with evaluation of estrous detection patches was performed three times daily beginning on day −2 and continued until follicle aspiration on day 1. Estrus was determined when cows stood to be mounted or had a fully activated patch. Ovaries of all cows were examined by an experienced technician using trans-rectal ultrasonography (Aloka 3500 with 7.5 MHz probe) on days −9, −2, 0, and 1. Any corpora lutea (CL) present and all follicles greater than 7 mm were recorded. Follicle size was defined as the average of the greatest diameter and the diameter perpendicular to it. Body weights and body condition score (BCS; scale of 1–9 in which 1 = emaciated and 9 = obese; [30]) were collected.

### 2.2. Follicle Classification

Animals were assigned to the following follicle classifications based on largest follicle size on day 0, estrous expression, and GnRH2 treatment: (1) Small (≤11.7 mm follicle, no estrous expression, and received GnRH2 injection), (2) Large (≥12.7 mm follicle, no estrous expression, and received GnRH2), and (3) Spontaneous (11.7–14 mm follicle, estrus expressed, and no GnRH2 injection; Figure 1b). The rationale for the establishment of the preceding groups was based on our previous studies in which postpartum beef cows that ovulated follicles <11.3 mm had a lower pregnancy rate following GnRH-induced ovulation and were less likely to support pregnancy to day 60 compared to cows that ovulated larger follicles in response to GnRH. In postpartum beef cows that spontaneously expressed estrus and ovulated, there was no effect of ovulatory follicle size on the establishment or maintenance of pregnancy [20]. Therefore, a range of dominant follicle sizes were included in the spontaneous follicle group. Cows that exhibited estrus and spontaneously ovulated experienced higher pregnancy rates following insemination than animals that did not express estrus and undergo GnRH-induced ovulation [20,31]. Therefore, the transcriptomes collected from the spontaneous follicle group should have represented transcriptomes of oocytes and corresponding cumulus cells from physiologically optimal follicles.

### 2.3. Transvaginal Aspiration for Collection of Cumulus Oocyte Complexes (COCs)

On day 1, approximately 23 h after GnRH2 injection or detection of estrus (mean = 23 h; range = 17–31 h), transvaginal aspiration of the largest follicle on either ovary was performed by 1 of 2 experienced technicians to collect the Cumulus-Oocyte complex (Figure 1a). All cows received a spinal block via injection of approximately 5 mL of 2% lidocaine into the spinal cord at the first intercoccygeal space of the tailhead. Next, the perineal area of each cow was cleaned of all contaminants. An ultrasound guided aspiration device containing an 18-gauge needle and a series of tubing to allow for ovum pickup and follicular flushing was positioned in the anterior vagina. The needle was gently pushed through the vaginal wall before being guided through the ovarian cortex and into the antrum of the dominant follicle. Follicular fluid was withdrawn into a clean 12 mL syringe. The syringe was replaced, and PBS-TL HEPES media was flushed into the collapsed follicle before being withdrawn into the second syringe. The follicle was flushed 3–4 times with all flushed media being collected. The needle was then withdrawn into the probe and removed from the cow. The tubing was flushed and the probe washed with a dilute chlorohexidine solution, sprayed with 70% ethanol, and rinsed with PBS-TL HEPES between each cow.

### 2.4. COC Recovery and Processing

Syringes containing follicular fluid and subsequent follicular flushes were divided into 4-well Petri plates and searched to find the COC. Once the COC was located, it was collected with 500 µL of surrounding follicular fluid or media and placed into a 2 mL RNAse free Eppendorf tube to be vortexed for 40 s. The COC and associated media were removed and placed into a smaller search plate filled with PVA-TL HEPES. The oocyte was separated from the surrounding cumulus cells by cutting away outlying cumulus cells with a clean pair of needles and repeatedly pipetting the COC up and down with a Cooke pipettor (170 micron tip) to denude the oocyte from most surrounding cumulus cells. The oocyte was subsequently placed into a small plate with PVA-TL HEPES containing 1000 units of hyaluronidase per ml. The oocyte was further pipetted to completely denude it of all surrounding cumulus cells. Fully denuded oocytes were placed into 1.5 mL RNAse free collection tubes containing 9 µL of lysis buffer (RNAqueous^®^ MicroKit; Ambion^®^; Foster City, CA, USA). Cumulus cells were collected into a 1.5 mL RNAse free collection tube and centrifuged for 3 min at 1500× *g* to pellet CC. The supernatant was removed and 90 µL of lysis buffer was added to the CC pellet. Both sample types were snap frozen in liquid nitrogen and stored at −80 °C until RNA extraction.

### 2.5. Statistical Analysis of Animal Data

Analysis of variance (ANOVA) was utilized to confirm a difference in follicular diameter among follicle classifications and determine any significant differences among follicle classifications in cow age, weight, BCS, days post-partum, interval from PGF to GnRH2, interval from GnRH2 or estrus to follicle aspiration, and interval from follicle aspiration to oocyte or cumulus cell freezing.

### 2.6. RNA Extraction, Library Preparation, and RNA Sequencing

Total RNA was extracted from pools of 4 oocytes or the CC corresponding to the same 4 oocytes (*n* = 17 pools of four oocytes or 17 corresponding CC) using the AllPrep^®^ DNA/RNA Micro Kit (Qiagen, Germantown, MD, USA) with on-column DNAse digestion following the manufacturer’s instructions. Due to low expected RNA quantity, we utilized the full volume of eluted RNA for reverse transcription and therefore did not quantify RNA from each pool before reverse transcription was performed. The mean concentration of RNA estimated in each oocyte or CC pool was 6.72 ng and 22.5 ng, respectively, based on spectrometry of practice samples (oocyte pools *n* = 4; CC *n* = 8) utilizing the NanoDrop 1000 Spectrophotometer (Thermo Fisher Scientific, Wilmington, DE, USA). Full RNA eluate per pool of 4 oocytes or corresponding CC samples was used as input material for the reverse transcription and amplification with the Ovation^®^ RNA-Seq System V2 (NuGen Technologies, Inc., San Carlos, CA, USA) following the manufacturer’s instructions. This approach allows for amplification to begin at both the 3’ end and randomly throughout the sample, allowing for reads to be evenly distributed throughout the amplified complementary DNA (cDNA) sample. Therefore, cDNAs from both polyadenylated and non-polyadenylated mRNA transcripts were amplified. Amplified cDNA was purified with the MinElute^®^ Reaction Cleanup Kit (Qiagen, Germantown, MD, USA) and eluted in 22 µL of Buffer EB. We quantified cDNA on a NanoDrop 1000 Spectrophotometer (Thermo Fisher Scientific, Wilmington, DE, USA) and determined an average of 4586.6 ng or 7449.0 ng of cDNA in amplified oocyte or CC pools, respectively. Oocyte pool 13 was removed from the workflow at this point due to an abnormally low cDNA concentration (499 ng). A total of 2000 ng of the amplified cDNA from each oocyte or CC pool was diluted with RNAse free water to reach a concentration of 80 ng/µL and underwent library preparation using the TruSeq DNA PCR-Free Library Preparation Kit (Illumina, Inc., San Diego, CA, USA). Sequencing was performed on an Illumina HiSeq 2000 (Illumina, Inc., San Diego, CA, USA) to produce 100 nucleotide long, single end reads. Raw sequences and count data are deposited in the Gene Expression Omnibus database (GSE176344).

### 2.7. RNA Sequencing Data Processing

Sequence adaptors were removed and remaining sequences were filtered for quality using fqtrim [32]. Sequences were retained if they met a median quality score of 25 and had a minimum read length of 30 base pairs. Filtered sequences were aligned to the Bos taurus reference genome UMD3.1 using Hisat2mapper [33]. FeatureCounts was used to quantify transcript abundance in each oocyte or cumulus cell pool using Bos taurus gene annotation from Ensembl [34,35].

### 2.8. Determination of Differentially Abundant Gene Transcripts

All analytical procedures were carried out utilizing R software, and the corresponding code is available as Appendix A [36]. We calculated counts per million (CPM) and filtered count data generated for oocyte and CC pools to retain only transcripts with at least 5 CPM in at least 4 oocyte pools or 5 CC pools. We then utilized the Bioconductor packages “edgeR” [37] and “DESeq2” [38] to perform differential gene expression analyses between oocyte or CC pools from small compared to large follicles, small compared to spontaneous follicles, and large compared to spontaneous follicles (Figure 1b). For each comparison, we inferred a transcript as differentially expressed if the nominal *p* value was ≤0.01 in both “edgeR” and “DESeq2” analyses. This nominal *p* value corresponded to an empirical false discovery rate (eFDR) *p* value ≤ 0.02 (Appendix A), which was calculated using 10,000 randomizations of sample reshuffling according to procedures and formulas outlined elsewhere [39,40,41,42,43]. Oocyte library 3 generated outlying transcript profiles for a number of differentially expressed transcripts that corresponded to significant pathways or gene ontologies during functional analysis in the small versus spontaneous follicle oocyte analysis (Appendix A). Therefore, we performed a second analysis of oocyte pool comparisons of small versus spontaneous follicles in which library 3 was removed from the dataset to validate the DEGs and functional analysis results. To maintain consistency, the same procedure was performed for the small versus large oocyte pool analysis. For these comparisons, transcripts were reported as DEGs only if they appeared as DEGs in both the original analysis and the validation analysis that removed oocyte library 3. We focus on transcripts and results of functional analyses of DEGs that remained following validation, but to maintain transparency, we also reported all DEGs, gene ontologies, and pathways identified in the original analysis in the Appendix A.

### 2.9. Gene Ontology and Pathway Enrichment Analyses

Differentially abundant transcripts from each comparison were uploaded into the Gene Ontology (GO) knowledgebase where we performed GO enrichment analyses of each DEG list compared against transcripts detected in oocytes or CC during our study as background to identify PANTHER pathways, REACTOME pathways, GO Biological Processes, GO Molecular Functions, or GO Cellular Components enriched with DEGs between each follicle classification in oocytes or CC [44,45,46]. We also uploaded each DEG list to the Database for Annotation, Visualization and Integrated Discovery (DAVID) knowledgebase to detect significantly enriched KEGG pathways, GO biological processes, GO molecular functions, or GO cellular components [47,48]. Enrichment of pathways or GO for DEGs was determined to be significant if the FDR < 0.05.

## 3. Results

### 3.1. Follicle Aspiration and Animal Data

A total of 127 COCs were recovered (51%) from 250 cows following transvaginal follicle aspiration. Of the 127 COCs collected, 87 COCs were allotted into small (*n* = 30), large (*n* = 36), and spontaneous (*n* = 21) follicle classifications. We aimed to generate RNA sequencing data in replicates of six libraries per follicle classification and cell type, and we used oocytes or CC from four COCs to prepare each library; therefore, in small and large follicle classifications, we selected samples with the most uniform parameters listed below to eliminate confounding effects of cow or collection timeline. Follicle diameter (mean ± SEM (range)) at GnRH2 injection differed among the small (10.4 ± 0.1a mm (8.5–11.7 mm)), large (13.6 ± 0.1b mm (12.7–15.3 mm)), and spontaneous follicle classifications (12.2 ± 0.2c mm (11.7–14.0 mm); *p* < 0.0001; Table 1). There was no difference among the oocyte and CC pools in cow age (*p* > 0.69), weight (*p* > 0.54), body condition score (*p* > 0.83), days postpartum (*p* > 0.70), time from PGF to GnRH2 injection (*p* > 0.89), time from GnRH2 injection or estrus to follicle aspiration (*p* > 0.06), and time from follicle aspiration to freezing of the oocyte (*p* > 0.57; Table 1).

### 3.2. Overview of RNA Sequencing Data

We generated transcriptome data from pools of four oocytes and their associated CC that originated from small (*n* = 6 pools for oocytes and CC), large (*n* = 6 pools for oocytes and CC), and spontaneous (*n* = 4 pools for oocytes; 5 pools for CC) follicle classifications (Figure 1b). Sequencing produced a total of 865,701,243 reads, with an average of ~23 million reads per pool for oocytes and ~29 million reads per pool for CC (Appendix A). We filtered aligned reads to eliminate genes with low expression and generated a list of 10,076 mRNA transcripts in oocyte pools and 9712 mRNA transcripts in CC pools to be utilized for differential gene transcript expression analyses. Of the above transcripts utilized for DEG analyses, 8083 were present within both oocyte and CC pools, whereas 1993 and 1629 transcripts were present in only oocyte or CC pools, respectively.

### 3.3. Differentially Abundant Gene Transcripts in Oocyte Pools

Analyses of differential gene transcript abundance in oocyte pools revealed 69 DEGs between oocyte pools from small versus large follicle classifications (32 upregulated and 37 downregulated in small follicles; eFDR < 0.02; Figure 2; Appendix A). We identified an additional 94 (31 upregulated and 63 downregulated in small follicles) and 83 (43 upregulated and 40 downregulated in large follicles) mRNA transcripts that displayed differential expression between small versus spontaneous and large versus spontaneous follicle classifications, respectively (eFDR < 0.02; Figure 2; Appendix A). Though 15, 4, and 12 transcripts were identified as DEGs in more than one comparison of follicle classifications, no transcripts were DEGs in all three comparisons.

Gene ontology and pathway analyses of the 94 DEGs between oocyte pools from small and spontaneous follicles revealed significant enrichment for the KEGG pathway “oxidative phosphorylation” (*ATP6V0D1*, mt*ND1*, mt*ND5*, *NDUFA7*, *NDUFA8*, mt*COX1*, *COX6A1*, *COX5B*; FDR < 0.01; Figure 2, Appendix A). Though no pathways or gene ontologies were significantly enriched with differentially expressed transcripts between oocyte pools for small versus large or large versus spontaneous follicle classifications, it is noteworthy to mention that five transcripts differentially expressed between large and spontaneous oocyte pools were related to oxidative phosphorylation (*ATP5G2*, *COX11*, *NDUFA13*, *NDUFA4*, *COX7A2*; Figure 2, Appendix A). Additionally, DEGs that were more abundant in oocytes from large compared to small follicles included *COX5A*, *MRPL48*, and *ATP6VID* (Figure 2, Appendix A), which are also involved in oxidative phosphorylation. Also of interest, the transcripts *FGF12* and protein kinase c (*PRKCG*) were DEGs in small versus large follicle oocyte pools (Figure 2, Appendix A).

### 3.4. Differentially Abundant Gene Transcripts in Cumulus Cell Pools

Differential gene transcript analysis of CC pools revealed 128 DEGs between CC pools from small versus large follicle classifications (30 upregulated and 98 downregulated in small follicles; eFDR < 0.02; Figure 3; Appendix A). An additional 98 (18 upregulated and 80 downregulated in small follicles) and 80 (12 upregulated and 68 downregulated in large follicles) mRNA transcripts with differences in abundance were identified between small versus spontaneous and large versus spontaneous follicle classifications, respectively (eFDR < 0.02; Figure 3; Appendix A). Though no transcripts were identified as DEGS in all three follicle classification comparisons, 8, 5, and 23 transcripts were identified as DEGs in two of the follicle classification comparisons.

Pathway analyses of the 128 DEGs between small and large CC pools revealed significant enrichment for the KEGG pathways “glycolysis” (*ALDOC*, *GPI*, *HK2*, *LDHA*, *PFKP*, *TRI1*; FDR < 0.01; Figure 3; Appendix A), “carbon metabolism” (*ALDOC*, *GPI*, *HK2*, *PFKP*, *SHMT2*, *TPI1*; FDR < 0.04), and “fructose/mannose metabolism” (*ALDOC*, *HK2*, *PFKP*, *TPI1*; FDR < 0.04; Figure 3; Appendix A). The PANTHER pathway “glycolysis” and multiple gene ontologies related to glycolytic processes were enriched with DEG between small and large follicle CC pools (Appendix A). Interestingly, all differentially expressed transcripts identified in these pathways were downregulated in CC pools from small follicles (Figure 3; Appendix A). Similarly, *HK2* was more highly expressed in cumulus cells from spontaneous compared to small follicles (Figure 3; Appendix A). Transcripts for two glucose transporters (*SLC2A1*, *SLC2A10*), a lactate transporter (*SLC16A3*), versican (*VCAN*) and *FGF11* were also upregulated in large vs. small CC pools (Figure 3; Appendix A). Additionally, the transcript encoding for *CoQ10A* was more highly expressed in cumulus cells of small versus large follicles (Appendix A), and *FGF2* was more highly expressed in cumulus cells from the spontaneous follicle group compared to both the small and large follicle groups (Figure 3; Appendix A). Though no pathways or gene ontologies were significantly enriched with differentially expressed transcripts between CC pools for small and spontaneous follicles, the KEGG pathway “lysosome” (*CD68*, *AP1S3*, *CTSZ*, *GLB1*, *GGA3*, *PSAP*; FDR < 0.03; Appendix A) was significantly enriched with transcripts from the 80 DEGs between CC pools from large and spontaneous follicle classifications.

## 4. Discussion

In postpartum beef cows, fertilization rate and embryo quality were decreased on day seven following GnRH-induced ovulation of small compared to large dominant follicles [18]. A GnRH-induced gonadotropin surge, in a cow with a physiologically immature follicle, may induce a premature stop to oocyte transcription and (or) transfer of mRNA from cumulus cells to the oocyte and thereby reduce the competence of an oocyte for fertilization and further embryonic development. The current experiment was conducted to test the hypothesis that the physiological status of an ovulatory follicle affects the bovine cumulus cell and (or) oocyte transcriptome.

### 4.1. Cumulus Cell Transcriptome

A major finding of this study was that the expression of glucose transporters and rate limiting glycolytic enzymes was less abundant in cumulus cells from small compared to large follicles following a GnRH-induced gonadotropin surge. Glucose can enter cumulus cells by facilitated transport via glucose transporters (GLUTs; [49]). Expression of transcripts encoding two glucose transporters (*SLC2A1* (GLUT1) and *SLC2A10* (GLUT 10)) was reduced in cumulus cells in the small versus large follicle groups. GLUT1 has been identified as a major glucose transporter in bovine follicular cells [50]. During in vitro maturation of bovine COCs, the addition of lysophosphatidic acid increased expression of GLUT1 and glucose uptake [51]. Treatment of murine COCs with inhibitors of the GLUT system decreased glucose uptake by cumulus cells and the enclosed oocyte [52]. Therefore, reduced transport of glucose into cumulus cells of small compared to large cumulus cells may result in reduced substrate availability for glycolysis.

Developmental competence of bovine oocytes was positively associated with increased glycolytic activity [53]. Cumulus cells have a high demand for glucose, which is converted by glycolysis to pyruvate, the primary energy source for the oocyte [54]. Bovine oocytes have decreased ability to take up glucose and have a low rate of glycolysis [55,56]. Oocytes depend on the surrounding cumulus cells for products of the glycolytic pathway (i.e., pyruvate and lactate) for oxidative metabolism [54,57]. Bovine oocytes have very low phosphofructokinase (*PFKP*) activity, which is the rate limiting enzyme in glycolysis [8]. Cumulus cells have greater *PFKP* expression than oocytes [8] and expression of PFKP was greater in cumulus cells surrounding mature versus immature human oocytes [58]. Supplementation of denuded bovine oocytes, in vitro, with pyruvate and lactate plus NAD+ promoted oocyte maturation [59].

In this study, the glycolytic pathway emerged as a major difference between the cumulus cell transcriptome of the small and large follicle groups. More specifically, expressions of six genes encoding glycolytic enzymes (*HK2*, *GPI*, *PFKP*, *ALDOC*, *TPI1*, and *LDHA*) in cumulus cells from small follicles were less abundant than in cumulus cells from large follicles. In addition, *HK2* expression was more abundant in cumulus cells of the spontaneous versus small follicle group. In mice, expression of glycolytic enzymes (PFKP and LDHA) was greater in cumulus cells compared to mural granulosa cells and removal of oocytes from cumulus cell oocyte complexes decreased glycolysis and the expression of the preceding enzymes [60,61]. In mice, the oocyte can increase the availability of pyruvate by upregulating glycolytic enzymes. Stimulation of glycolysis in cumulus cells by the murine oocyte is via paracrine action (i.e., BMP 15 and FGFs; [62]). However, in this study, there was no difference in expression of *BMP 15* in oocytes in the small and large follicle groups. Although *FGF12* expression was increased in oocytes from the small compared to the large follicle group, *FGF12* is reported to have a nuclear localization and is not secreted [63].

Pyruvate, a product of the glycolytic pathway, can enter the Krebs cycle or be converted to lactate via lactate dehydrogenase A (*LDHA*). The enzyme LDHA converts pyruvate to lactate, thereby generating NAD+, which is required for sustained glycolytic activity [64]. A decline in murine oocyte quality occurred during reproductive aging and was associated with a reduction in NAD+. However, supplementation with an NAD+ precursor, nicotinamide mononucleotide, in the water of aged mice improved oocyte quality and restored fertility [65]. In the current study, transcripts encoding for LDHA and a lactate exporter (*SLC16A3*; Monocarboxylic Acid Transporter 4 (MCT4)) were more highly expressed by cumulus cells in the large versus small follicle group. MCT4 is expressed in cells that have a high rate of glycolysis and MCT4 can be induced by hypoxia [66]. Production and export of lactate may promote oocyte competence since addition of lactate and NAD+ to maturation medium promoted competence in denuded bovine oocytes [59]. In the bovine oocyte, LDHB is the primary lactate dehydrogenase isoenzyme present [59] and, in this study, *LDHB* was expressed in oocyte pools from each of the follicle classification groups. Therefore, lactate may be produced and exported at a higher rate from cumulus cells of large compared to small follicles and converted within oocytes to pyruvate to provide substrate for oxidative phosphorylation.

An impaired glycolytic pathway in cumulus cells of small compared to large dominant follicles may result in reduced pyruvate availability for ATP production via the Krebs cycle in oocytes. Mammalian oocytes require an adequate amount of ATP to successfully complete spindle formation, chromatid separation, and cleavage divisions [67]. ATP content is an important marker of oocyte competence [68,69] and reduced ATP availability in oocytes may contribute to reduced fertilization success and decreased embryo quality observed after GnRH-induced ovulation of small compared to large bovine follicles [18].

In addition to glycolysis, glucose metabolism occurs within cumulus cells via the pentose phosphate pathway and hexosamine pathway [54]. In this study, enzymes associated with the pentose phosphate pathway in cumulus cells were not differentially expressed among any of the groups. The hexosamine pathway results in production of hyaluronic acid, the primary component of the cumulus matrix in cattle. Another component of the cumulus matrix is versican (*VCAN*), which is induced by the pre-ovulatory gonadotropin surge. Versican is a proteoglycan that binds to hyaluronic acid during cumulus expansion to help stabilize the cumulus matrix. *VCAN* was upregulated in cumulus cells from large follicles compared to small follicles in the present study. An increase in *VCAN* expression in human cumulus cells was positively associated with early embryo morphology score [58] and embryo quality [70]. In rodents, cumulus expansion is reported to affect oocyte maturation [71] and fertilization [72]; however, in cattle this relationship is less clear. Gutinisky et al. (2007) reported in cattle that cumulus expansion, in vitro, was not associated with oocyte nuclear maturation, but facilitated sperm penetration and fertilization [73]. Furthermore, following fertilization, oocyte developmental competence was not dependent upon cumulus expansion.

In mammals, members of the fibroblast growth factor family frequently act via paracrine mechanisms to promote embryonic development [74]. In the present study, fibroblast growth factor 2 (*FGF2*) expression was more abundant in the cumulus cells from the spontaneous follicle group compared to both the small and large follicle groups. The FGF2 receptor (*FGFR1*) was expressed in the transcriptome of both cumulus cells and oocytes. In cattle, inhibition of FGF signaling on the day of IVF inhibited blastocyst formation, whereas supplementation of FGF2 following IVF increased blastocyst formation [75]. In mice, FGF2 was reported to have an autocrine/paracrine action to promote cumulus expansion and oocyte meiotic maturation [76]. Although *FGF 11* expression was more abundant in cumulus cells of large compared to small follicles, *FGF 11* is not secreted and its biological function in the cumulus cell/oocyte complex is not currently known [63]. However, *FGF11* expression was upregulated in bovine cumulus cells from larger (> 8mm) compared to smaller (1–3 mm) follicles and proposed to be a marker of oocyte competence [77].

### 4.2. Oocyte Transcriptome

The most prominent result from functional analysis of DEGs among oocyte pools was enrichment of the KEGG pathway “oxidative phosphorylation” with DEGs in oocyte pools from the small versus spontaneous follicle classifications. Mitochondrial and nuclear transcripts encoding for subunits of oxidative phosphorylation complex I (NADH: ubiquinone oxidoreductase) and complex IV (cytochrome c oxidase) were differentially expressed in small compared to spontaneous oocyte pools. Interestingly, additional DEGs associated with oxidative phosphorylation or mitochondrial function were identified among large versus spontaneous and small versus large follicle oocyte comparisons. The oocyte’s dependance on mitochondrial ATP production increases drastically during oocyte maturation, and mitochondrial number and ATP concentration have been demonstrated to increase during oocyte maturation [69,78,79]. Oocyte mitochondrial production of ATP must support metabolic needs of meiotic resumption, fertilization, and pre-blastocyst embryo development [8,67,80,81]. Numerous studies have demonstrated the relationship between oocyte mitochondrial number, indicated by mitochondrial DNA copy number, and oocyte developmental competence for fertility in cattle and other mammalian species [82,83,84,85,86]. Reduced capacity for oxidative phosphorylation could have contributed to the reduced fertilization and embryo quality in oocytes from small follicles of a previous study [18].

Though almost 100% of nuclear mRNA DEGs from each comparison were upregulated in the follicle classifications associated with presumably higher oocyte developmental competence (spontaneous versus small or large follicles and large versus small follicles), it is unknown why a small number of transcripts derived from the mitochondrial genome (*ND1*, *ND5*, *COX1*) were upregulated in oocyte pools from small compared to spontaneous follicles. The mitochondrial genome encodes only 13 protein coding genes, alongside the translational machinery required to assemble mitochondrial proteins [87]. Because of the intimate linkage of the mitochondrial genome/function with oxidative phosphorylation, the increased abundance of *ND1*, *ND5*, and *COX1* transcripts in small follicle oocyte pools may be physiologically significant. Mitochondrial mRNA abundance is directly linked to the production of the oxidative phosphorylation complexes and is mainly controlled post-transcriptionally to ensure a steady-state level of mitochondrial encoded proteins [87]. Because of the dynamic, and poorly understood, mechanisms of mitochondrial mRNA transcription and transcript stability [88], the biological significance of the increase in mitochondrial mRNA transcripts associated with oocyte oxidative phosphorylation requires further investigation.

In addition to transcripts related to oxidative phosphorylation, expression of protein kinase C, gamma (*PRKCG*) was less abundant in oocytes from small compared to large follicles. Protein kinase C, gamma is a member of the classical protein kinase C subclass (cPKC), which is the most studied category of PKC in the oocyte. This subclass is known to have an important role in oocyte maturation, meiotic resumption, and fertilization [89,90,91]. Protein kinase C modulates intracellular calcium oscillations, within the oocyte, following fertilization, leading to cortical granule release and a subsequent polyspermy block [90,91,92]. Calcium oscillations at the time of fertilization have also been demonstrated to increase mitochondrial ATP production to support metabolic requirements of fertilization [93].

### 4.3. Cumulus Cell and Oocyte Markers of Oocyte Competence

Numerous studies have aimed to identify specific markers of oocyte competence in a variety of mammalian species [26,27,28]. Previous studies in cattle reported that increased expression of follistatin (*FST*), inhibin βa (*INHA*), and inhibin βb (*INHB*) in oocytes collected from 3 to 7 mm follicles was associated with increased oocyte competence based on two experimental models (comparison of oocytes from prepubertal versus adult ovaries and comparison of early versus late cleaving embryos; [28]). In the current study, transcript abundance for *FST*, *INHA*, and *INHB* was similar for the three follicle classification groups. This is not entirely surprising, since Cumulus-Oocyte complexes in the present study were collected from follicles that were larger than those reported by Patel et al. (2007), and in the current experiment, all samples experienced expansion of the cumulus cells. Oocytes collected in the current study were from dominant follicles compared to prepubertal or slaughterhouse ovaries. Therefore, one explanation for the differences in gene expression could be the result of different stage oocytes in the two studies.

Efforts have also been directed at identifying cumulus cell markers of oocyte competence [22,23,25]. Bettegowda et al. (2008) reported that *CTSB*, *CTSS*, and *CTSZ* expression was higher in cumulus cells from prepubertal bovine cumulus cells (less competent oocytes) compared to cumulus cells (high competent oocytes) from mature cows [22]. There have also been reports that inhibiting cathepsin activity during in vitro cultures leads to an increase in blastocyst development [22,29]. In the preceding study, oocytes with increased blastocyst rate also had lower cumulus cell expression of *CTSB*, *CTSS*, and *CTSZ*. Pohler (2011) reported that as dominant follicular diameter increased (from 9.5 to 14 mm), there was a decrease in *CTSB* expression and there tended to be a decrease in *CTSZ* expression in cumulus cells [94]. Collectively, these data suggest that smaller ovulatory follicles may contain oocytes that are less competent based on cathepsin expression. However, in the current study, there was no difference in cumulus cell *CTSZ* expression between small and large follicles or between small and spontaneous follicles. Surprisingly, *CTSZ* expression was downregulated in cumulus cells of large compared to spontaneous follicles. Expression of *FGF11* was reported to be a marker of oocyte competence [77] and in the current study was more highly expressed in cumulus cells of large compared to small follicles. However, expression of other reported bovine cumulus cell markers (*ARHGAP22*, *COL18A1*, *GPC4*, *IGFBP4*, and *SPRY1*; [77]) associated with oocyte competence in in vitro systems was not different among the small, large, and spontaneous follicle groups.

## 5. Conclusions

In summary, analysis of the transcriptome of bovine cumulus cells and oocytes from dominant follicles that differ in size (small versus large) or physiological status (estrous expression versus no estrous expression) revealed differentially abundant transcripts that were associated with important pathways leading to the acquisition of oocyte competence. The most resounding differences identified among oocyte or CC pools was reduced expression of transcripts encoding for glycolytic enzymes in CC from small versus large follicles. Furthermore, reduced expression of glucose transporters in small versus large CC pools and differential expression of transcripts related to oxidative phosphorylation in small and large versus spontaneous follicle oocytes indicates further metabolic impact when cows are induced to ovulate a physiologically suboptimal follicle. Though functional analyses of DEGs revealed the strong differences discussed above between small and large or spontaneous follicles, the DEGs between large and spontaneous follicles were not as strongly represented in significant biological processes or pathways. These results support previous studies by our research team that did not detect reduced pregnancy rates or embryonic mortality between large follicle induced to ovulate or follicles that experienced spontaneous ovulation following estrus [20]. Such results suggest that cumulus oocyte complexes from large follicles have likely passed a threshold of metabolic competence required for subsequent fertilization and embryo development. Future studies are necessary to further understand the altered metabolic capacity of small follicle COCs and determine potential mechanisms to improve the competency of small follicle oocytes. Improvements in assisted reproductive procedures to result in increased follicle size before ovulation during fixed-time artificial insemination or oocyte collection for in vitro techniques are practical applications to improve reduced oocyte competence in smaller follicles.

## Figures and Tables

**Figure 1 genes-12-00893-f001:**
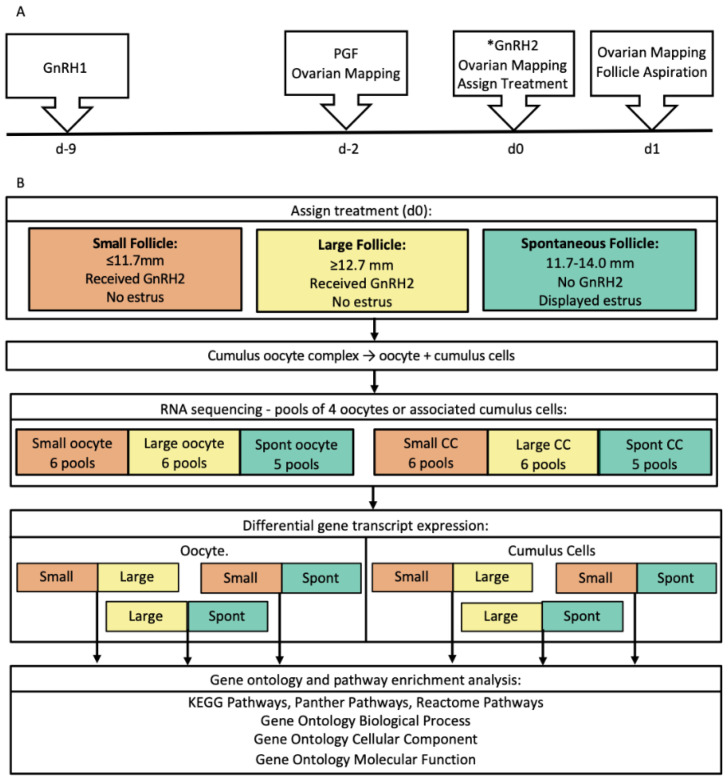
(**A**) Protocol for synchronization of ovulation and determination of follicle diameter by ultrasonography. The cumulus oocyte complex was collected by transvaginal aspiration on d 1. * Cows assigned to the Spontaneous (Spont) follicle classification did not receive GnRH2 injection on day 0. (**B**) Cows that did not express estrus by d 0 were assigned to the Small (≤11.7 mm) or Large Follicle (≥12.7 mm) groups based on largest follicle diameter on d 0. Cows that expressed estrus by d 0 were assigned to the Spont Follicle group regardless of ovulatory follicle size. Pools of four oocytes or associated cumulus cells were prepared for generation of RNA sequencing data and analyses. Differential gene transcript expression for both oocyte and cumulus cell pools was analyzed as follows: Small vs. Large, Small vs. Spont, and Large vs. Spont. Gene ontology and pathway enrichment was analyzed as shown in the figure.

**Figure 2 genes-12-00893-f002:**
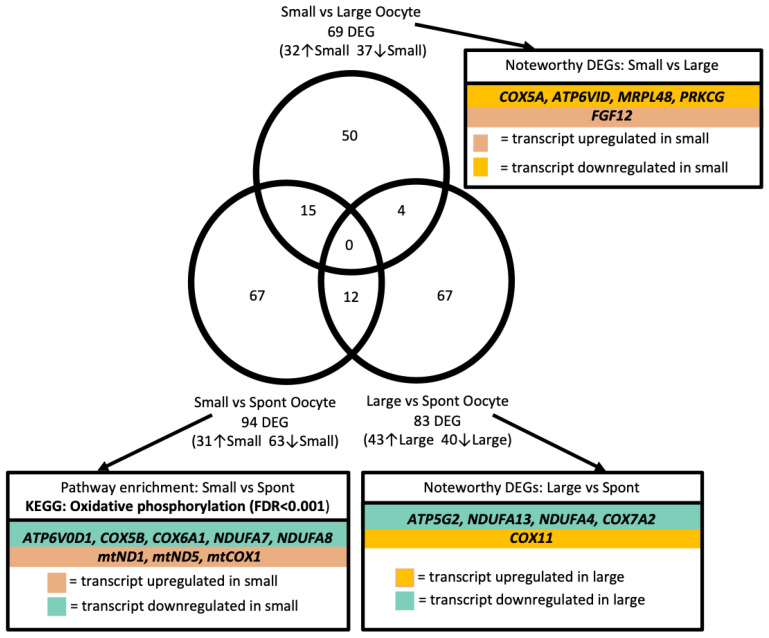
Differential gene transcript abundance among oocyte pools. Venn diagram displaying 69, 94, and 83 differentially abundant gene transcripts (DEGs) in comparisons of Small vs. Large, Small vs. Spontaneous (Spont), and Large vs. Spont oocyte pools, respectively. A total of 50, 67, and 67 DEGs were identified in only Small vs. Large, Small vs. Spont, or Large vs. Spont oocyte comparisons, respectively. A total of 15 transcripts were identified as DEGs in both Small vs. Large and Small vs. Spont comparisons; 4 transcripts were identified as DEGs in both Small vs. Large and Large vs. Spont comparisons; and 12 transcripts were identified in both Small vs. Spont and Large vs. Spont comparisons. No transcripts appeared as DEGs in all three comparisons of follicle classifications. The KEGG pathway significantly enriched with DEGs, alongside additional noteworthy DEGs, are indicated for each comparison.

**Figure 3 genes-12-00893-f003:**
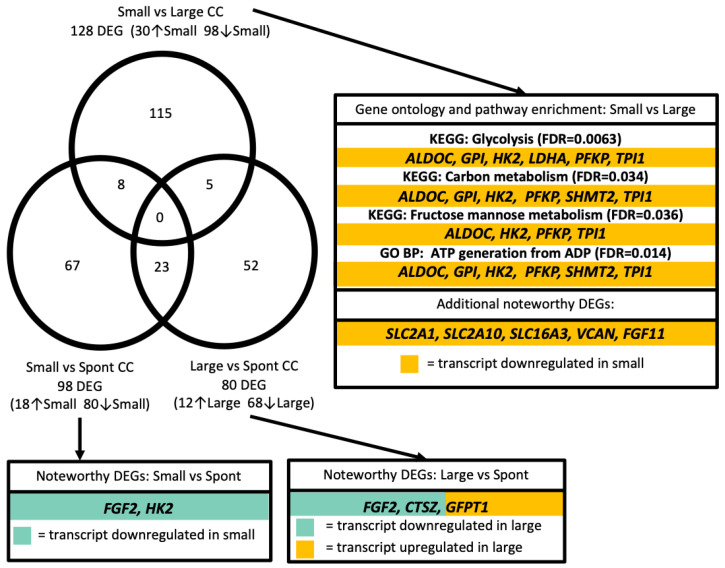
Differential gene transcript abundance among cumulus cell (CC) pools. Venn diagram displaying 128, 98, and 80 differentially abundant gene transcripts (DEGs) in comparisons of Small vs. Large, Small vs. Spontaneous (Spont), and Large vs. Spont CC pools, respectively. A total of 115, 67, and 52 DEGs were identified in only Small vs. Large, Small vs. Spont, or Large vs. Spont CC comparisons, respectively. A total of 8 transcripts were identified as DEGs in both Small vs. Large and Small vs. Spont comparisons; 5 transcripts were identified as DEGs in both Small vs. Large and Large vs. Spont comparisons; and 23 transcripts were identified in both Small vs. Spont and Large vs. Spont comparisons. No transcripts appeared as DEGs in all three comparisons of follicle classifications. Example gene ontology (GO) and KEGG pathways significantly enriched with DEGs, alongside additional noteworthy DEGs, are indicated for each comparison.

**Table 1 genes-12-00893-t001:** Parameters associated with the collection of oocyte and cumulus cell pools for small, large, or spontaneous follicles.

Parameter	Small Follicle Classification	Large Follicle Classification	Spontaneous Follicle Classification
Follicle size at GnRH2 ^a^	10.4 ^x^ ± 0.1 mm (8.5–11.7 mm)	13.6 ^y^ ± 0.1 mm (12.7–15.3 mm)	12.2 ^z^ ± 0.2 mm (11.7–14.0 mm)
Cow age ^b^	6.5 ± 0.4 years (4–12 years)	6.3 ± 0.4 years (4–9 years)	6.9 ± 0.5 years (4–13 years)
Cow weight ^c^	548 ± 11 kg (454–674 kg)	564 ± 11 kg (452–668 kg)	548 ± 14 kg (468–646 kg)
Cow BCS ^d^	4.8 ± 0.1 (4–5)	4.8 ± 0.1 (4–6)	4.7 ± 0.1 (4–5)
Cow days postpartum ^e^	88 ± 1.7 days (65–96 days)	86 ± 1.7 days (58–98 days)	86 ± 2.0 days (76–95 days)
Time from PGF to GnRH2 ^f^	51 ± 1.5 h (43–56 h)	51 ± 1.5 h (43–56 h)	NA
Time from GnRH2 or estrous onset to follicle aspiration ^g^	23 ± 0.8 h (18–31 h)	22 ± 0.6 h (17–30 h)	21 ± 0.7 h (17–24 h)
Time from follicle aspiration to snap freezing of samples ^h^	22 ± 1.6 min (11–45 min)	24 ± 1.6 min (11–34 min)	25 ± 2.0 min (12–45 min)

^a^ Size of the pre-ovulatory follicle on day 0 at GnRH2 injection (^xyz^
*p* < 0.0001), Mean ± SEM (range); ^b^ Cow age, Mean ± SEM (range); ^c^ Body weight, Mean ± SEM (range); ^d^ Body condition score (BCS), Mean ± SEM (range); ^e^ Days postpartum, Mean ± SEM (range); ^f^ Time from injection of PGF to injection of GnRH2 in cows within the small or large follicle size classifications, Mean ± SEM (range); NA = not applicable; ^g^ Time from injection of GnRH2 to follicle aspiration in cows within the small or large follicle size classifications, Mean ± SEM (range); NA = not applicable; ^h^ Time from follicle aspiration to snap freezing of the oocyte and cumulus cells, Mean ± SEM (range).

## Data Availability

Publicly available datasets were analyzed in this study. This data can be found here: Gene Expression Omnibus database (GSE176344; https://www.ncbi.nlm.nih.gov/geo/query/acc.cgi?acc=GSE176344, accessed on 8 June 2021).

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
