# Peer review of "Differential Transcript Profiles in Cumulus-Oocyte Complexes Originating from Pre-Ovulatory Follicles of Varied Physiological Maturity in Beef Cows"

_genes, 2021, doi:10.3390/genes12060893_

Round 1
Reviewer 1 Report
It seems to me that the verb of the first sentence in the abstract is misleading, and perhaps should read "decreases" instead.
Line 251 makes reference to "the manuscript", probably refering to the submitted paper, but I find it inadecuate.
The Venn diagrams are certainly compeling but the inner zeroes get no attention whatsoever.
Upregulated DEGs in suboptimal oocytes and cumuli get very limited comment.
Conclusions are too brief for such large set of results and, further, intercellular transfer of mRNA is scantly commented despite its prominent mention in the introduction.
Author Response
We appreciate your time reviewing our manuscript and have made edits in response to your suggestions outlined below. We performed a complete spell check and made minor changes to this area as well. Please see all edits in track changes of the uploaded revised manuscript.
It seems to me that the verb of the first sentence in the abstract is misleading, and perhaps should read "decreases" instead. We appreciate the reviewer’s comment but believe “decreased” to be the correct tense because this specific sentence/word choice is related to results previously published.
Line 251 makes reference to "the manuscript", probably refering to the submitted paper, but I find it inadecuate.Thank you for suggesting this edit. We removed the words “the manuscript” from this sentence.
The Venn diagrams are certainly compelling but the inner zeroes get no attention whatsoever. We appreciate the reviewer’s comment and added text in the main body and figure legends to address this area.
Upregulated DEGs in suboptimal oocytes and cumuli get very limited comment. We agree that upregulated DEGS in the small follicle classification received limited comment. We individually explored each DEG in our dataset, and few transcripts that were not discussed during commentary on functional analysis had clear biological relevance. We did discuss the increased expression of FGF12 in small follicle oocytes and ND2/ND6 in small follicle CC. These transcripts seemed most reasonable to mention, and we did not explicitly discuss other transcripts increased in small follicles because their relevance to oocyte competence was less clear.
Conclusions are too brief for such large set of results and, further, intercellular transfer of mRNA is scantly commented despite its prominent mention in the introduction. We appreciate the opportunity to improve the conclusions section of our manuscript. We added additional text related to the functional implications of few biologically relevant differences between large and spontaneous follicles, and we discussed practical implications of our work. We did not discuss intercellular transfer of mRNA in the discussion or conclusion because the scope of our work does not allow us to infer functional significance of this in our specific dataset or determine if specific transcripts were transferred from the CC or accumulated solely by the oocyte.
Reviewer 2 Report
This manuscript by Moorey et al. stems from the observation that induced ovulation of bovine small dominant follicles was associated with reduced pregnancy and increased embryonic mortality compared to induced ovulation of large follicles. The authors used RNA-seq of oocyte and associated cumulus cell pools to probe for differential gene expression between either small or large dominant follicles that were induced to ovulate, and also comparing small/large follicles induced to ovulate with follicles that ovulated spontaneously. They found that genes and pathways associated with glucose transport and glycolysis were expressed at lower levels in cumulus cells of small vs. large follicles induced to ovulate. Furthermore, genes associated with oxidative phosphorylation were generally expressed at higher levels in oocytes from large and spontaneously ovulating follicles. These data provide solid clues for understanding the mechanisms behind developmental deficiencies in the cumulus cells and oocytes of small dominant follicles induced to ovulate. I only have a couple of minor comments:
-- For readers unfamiliar with ovarian biology, perhaps have a sentence or two explaining the definition and biological context of the dominant follicle in the Introduction.
-- This is not critical, but it might be easier to follow if the oocyte/cumulus data or discussion are presented in the same order in all sections.
This is a study with relatively limited scope and clear presentation. In looking again, I realized that I have one more question for the authors. To the authors: It is curious to me that the numbers of differentially expressed genes (and their magnitudes) between the large and spontaneous groups are just as robust as those between the small and large groups. Yet there appear to be no functional/phenotypic differences between the large and spontaneous groups. If many of the differentially expressed genes in all groups relate to oxidative phosphorylation, with higher expression overall from small to large to spontaneous, does that mean that there is a certain threshold of oxidative phosphorylation gene expression that must be met for normal development? Or are some of the oxidative phosphorylation genes that have low expression in the small group just more functionally important than those that might be low in the large group?Author Response
Thank you for your attentive review and the helpful suggestions to improve our manuscript. We addressed your comments below, and we performed a complete spelling/grammar check to improve the manuscript. Please see the revised manuscript that has all changes tracked for your ease of review.
For readers unfamiliar with ovarian biology, perhaps have a sentence or two explaining the definition and biological context of the dominant follicle in the Introduction. Thank you for this excellent suggestion. We added/edited the following text in lines 64-71 to further explain the importance of the dominant follicle. “Female cattle recruit two to three waves of multiple antral follicles for growth during the estrous cycle, and one follicle becomes dominant during each follicular wave [6]. While progesterone levels are high, the dominant follicle cannot continue growth and will regress to allow the onset of a new follicular wave. The dominant follicle from the second or third follicular wave continues growth during the reduced progesterone levels that follow luteolysis. This follicle ovulates following the pre-ovulatory gonadotropin surge, and its physiological status near the time of ovulation may affect establishment and maintenance of pregnancy by affecting oocyte competence and the maternal environment [18, 19].”
This is not critical, but it might be easier to follow if the oocyte/cumulus data or discussion are presented in the same order in all sections. We appreciate your comment and note that this is not critical. We discussed this conundrum when constructing the manuscript, and after much thought decided to utilize the order submitted. We chose the order of oocyte and then cumulus in the results section because the oocyte is the gamete/likely of highest interest to the reader. Once we reached the discussion section of the manuscript, we switched the order because cumulus related discussion points were necessary to lay a foundation for the oocyte discussion and we felt the CC discussion was most robust/of greatest interest to the reader.
This is a study with relatively limited scope and clear presentation. In looking again, I realized that I have one more question for the authors. To the authors: It is curious to me that the numbers of differentially expressed genes (and their magnitudes) between the large and spontaneous groups are just as robust as those between the small and large groups. Yet there appear to be no functional/phenotypic differences between the large and spontaneous groups. If many of the differentially expressed genes in all groups relate to oxidative phosphorylation, with higher expression overall from small to large to spontaneous, does that mean that there is a certain threshold of oxidative phosphorylation gene expression that must be met for normal development? Or are some of the oxidative phosphorylation genes that have low expression in the small group just more functionally important than those that might be low in the large group? We were also intrigued by this discovery and agree that the reviewer's initial statements may be correct. We added text in lines 567-584 to briefly provide this rationale. "Though functional analyses of DEGs revealed the strong differences discussed above between small and large or spontaneous follicles, the DEGs between large and spontaneous follicles were not as strongly represented in significant biological processes or pathways. These results support previous studies by our research team that did not detect reduced pregnancy rates or embryonic mortality between large follicle induced to ovulate or follicles that experienced spontaneous ovulation following estrus [20]. Such results suggest that cumulus oocyte complexes from large follicles have likely passed a threshold of metabolic competence required for subsequent fertilization and embryo development."
Reviewer 3 Report
The manuscript described well the results of an elegant and nice study. I only miss a small paragraph at the end of discussion or at conclusions addressing practical implications.
Author Response
We appreciate your time and positive review of our manuscript. We performed a spelling/grammar check to improve the manuscript and addressed your comment below. Please see the revised version of the manuscript with all changes tracked for your ease of review.
The manuscript described well the results of an elegant and nice study. I only miss a small paragraph at the end of discussion or at conclusions addressing practical implications. We appreciate the opportunity to further expand this section of the manuscript. We added the following text to lines 584-590: “Future studies are necessary to further understand the altered metabolic capacity of small follicle COCs and determine potential mechanisms to improve competency of small follicle oocytes. Improvements in assisted reproductive procedures to result in increased follicle size before ovulation during fixed-time artificial insemination or oocyte collection for in vitro techniques are practical applications to improve reduced oocyte competence in smaller follicles.”